# Cell-Molecular Interactions of Nano- and Microparticles in Dental Implantology

**DOI:** 10.3390/ijms24032267

**Published:** 2023-01-23

**Authors:** Varvara Labis, Ernest Bazikyan, Denis Demin, Irina Dyachkova, Denis Zolotov, Alexey Volkov, Victor Asadchikov, Olga Zhigalina, Dmitry Khmelenin, Daria Kuptsova, Svetlana Petrichuk, Elena Semikina, Svetlana Sizova, Vladimir Oleinikov, Sergey Khaidukov, Ivan Kozlov

**Affiliations:** 1Stomatology Faculty, A.I. Yevdokimov Moscow State University of Medicine and Dentistry, 127473 Moscow, Russia; 2Center for Precision Genome Editing and Genetic Technologies for Biomedicine, Engelhardt Institute of Molecular Biology, Russian Academy of Sciences, 119991 Moscow, Russia; 3Federal Scientific Research Centre “Crystallography and Photonics” Russian Academy of Sciences, 119333 Moscow, Russia; 4Federal State Budgetary Institution “National Medical Research Center for Traumatology and Orthopedics Named after N.N. Priorov” of the Ministry of Health of the Russian Federation, 127299 Moscow, Russia; 5Department of Pathological Anatomy, Peoples’ Friendship University of Russia (RUDN University), 117198 Moscow, Russia; 6Department of Machine-Building Technologies, Bauman Moscow State Technical University, 105005 Moscow, Russia; 7Federal State Autonomous Institution “National Medical Research Center for Children’s Health”, Ministry of Health of the Russian Federation, 119991 Moscow, Russia; 8Department of Biomaterials and Bionanotechnology, Shemyakin & Ovchinnikov Institute of Bioorganic Chemistry of the Russian Academy of Sciences, 117997 Moscow, Russia; 9Institute of Professional Education, I.M. Sechenov First Moscow State Medical University, 119991 Moscow, Russia

**Keywords:** soft tissue biopsy specimens, supernatants with nanoscale metallic particles (NSMP), microscale particles, dental implants, X-ray microtomography (XMCT), X-ray fluorescence analysis (XRF), scanning electron microscopy (SEM), histological studies, cytometry, immunopathological inflammation

## Abstract

The role of metallic nano- and microparticles in the development of inflammation has not yet been investigated. Soft tissue biopsy specimens of the bone bed taken during surgical revisions, as well as supernatants obtained from the surface of the orthopedic structures and dental implants (control), were examined. Investigations were performed using X-ray microtomography, X-ray fluorescence analysis, and scanning electron microscopy. Histological studies of the bone bed tissues were performed. Nanoscale and microscale metallic particles were identified as participants in the inflammatory process in tissues. Supernatants containing nanoscale particles were obtained from the surfaces of 20 units of new dental implants. Early and late apoptosis and necrosis of immunocompetent cells after co-culture and induction by lipopolysaccharide and human venous blood serum were studied in an experiment with staging on the THP-1 (human monocytic) cell line using visualizing cytometry. As a result, it was found that nano- and microparticles emitted from the surface of the oxide layer of medical devices impregnated soft tissue biopsy specimens. By using different methods to analyze the cell–molecule interactions of nano- and microparticles both from a clinical perspective and an experimental research perspective, the possibility of forming a chronic immunopathological endogenous inflammatory process with an autoimmune component in the tissues was revealed.

## 1. Introduction

The clinical cases of mucositis and peri-implantitis in the practice of a dental surgeon, leading to the loss of dental implants, is the motivation for searching for the causes of etiopathogenesis of inflammatory complications [1,2]. Approaches to the study and the development of treatment methods for complications in dental implantology are more often aimed at levelling the clinical symptomatology of inflammation without attempting to study the mechanisms of its occurrence in tissues at the molecular, nanoscale, and microscale levels.

Close attention is paid to the bacterial factor in the development of mucositis and peri-implantitis [3,4].

Initially, the process of osseointegration of dental implants was justified by the inability of the immune system to recognize a foreign body made of a metal alloy based on titanium oxide [5], which was considered “bioinert” [6]. Later there was evidence of the possibility of developing a personalized reaction to the components of the titanium oxide-based alloy [7]. In 2011, immunology discovered pattern recognition receptors (PRRs) are involved in the recognition of all types of antigenic determinants [8] and began to study the interaction of these receptors with metals [9].

In the last decade, there has been an increase in the number of investigations aimed at studying the effects of metal oxide nanoparticles, such as TiO_2_ and ZnO, on animals and humans [9,10,11,12,13,14,15,16,17]. Various changes on the part of the immune system cells in response to nanoparticles in prepared supernatants based on metal oxides have been identified and tracked in works [10,11,12,13,14,15,16,17].

In [18], it was found that when a foreign material enters the soft tissues, an immune response aimed to organize the Foreign Body Response (FBR) occurs, which further leads to the formation of collagen encapsulation. FBR initiation includes macrophage activation leading to the formation of giant cells, fibroblast activation, and collagen matrix deposition. With the development of an acute inflammatory reaction in the tissues where the foreign material is introduced, the formation of Nod-like receptor protein 3 (NLRP3) and ASC (Apoptosis-associated speck-like protein containing a CARD) inflammasomes and the activation of caspase-1 and other participants of the immune response occur. At the same time, the authors note that caspase-1 and ASC play a priority role in the development of the fibrous capsule, but not NLRP3.

In addition to TiO_2_-based alloys, CoCr-based alloys, which may include Ni2+ as solder, are used for orthopedic structures that are fixed on dental implants as orthopedic supports.

In 2010, an inhibitory effect on TLR4 (toll-like receptor 4, CD284) in the pathogenesis of a delayed-type hypersensitivity reaction to Ni2+ was shown in a mouse model [19]. When Ni2+ enters the body for the first time, this hapten is presented to dendritic cells, which, in turn, leads to the formation of a T-cell immune response. The mechanisms of proinflammatory signal activation in the cell are still unknown. It is known only that proinflammatory cytokine genes are expressed through direct TLR activation without additional co-stimulation [19].

In dental implantology, there is a term “fibrosteointegration” introduced by P.I. Branemark, which means a negative outcome of dental implant engraftment. The presence of a connective tissue layer between the bone itself and the implant consisting of collagen fibers and coarse fibrous connective tissue determines the negative outcome of osseointegration. In this regard, the interpretation of the mechanism of engraftment of the dental implant made on the basis of TiO_2_ from the position of modern immunology becomes significant [20].

In the present work, using physical, histological, and immunological research methods, the role of nano- and microscale particles as etiopathogenetic participants in the processes of osseointegration and disintegration of dental implants was studied. The role of immunocompetent cells in response to nanoscale metallic particles (NSMPs) obtained from the surfaces of dental implants of two systems, Nobel Replace and Alpha Bio, in the supernatants, namely, in the development of peri-implantitis, was studied using an in vitro experimental investigation.

## 2. Results

### 2.1. Results of X-Ray Studies

Figure 1 shows the XMCT results of soft tissue biopsy specimens obtained from the bone bed zones in the area of four spontaneously disintegrated dental implants of the Alpha Gate (Alpha Gate Mazen Ltd., Kfar Qari, Israel) system and a control specimen in the area of a tooth with chronic inflammatory periodontal disease. As can be seen from the reconstructed images, the samples under study contain irregularly located foreign microinclusions of different sizes and densities.

Processing and analysis of the obtained three-dimensional reconstructions allowed us to estimate the location, sizes (equivalent diameter), linear absorption coefficients, and concentration of registered foreign inclusions in biological tissues. The equivalent diameter of the inclusions *EqD* was calculated according to the following Equation:(1)$$EqD=\sqrt[{3}]{\frac{6\times Volume}{\pi }}$$
where *Volume* is the volume of inclusion.

The concentration of foreign inclusions in the studied volume of the gingival part was estimated according to the following formula:(2)$$C_{B}=\frac{N_{B}}{V}$$
where *N_B_* is the number of particles and *V* is the volume of the studied sample of biological tissue.

Foreign microinclusions with sizes from ~17 to 560 μm were revealed (Figure 1b,d,f,h) with linear absorption coefficients from ~0.2 to 6.8 mm^−1^ (Figure 1a,c,e,g, contrast in the absorption coefficient from green to red). All inclusions can be separated mostly into two types—small (tens of microns), strongly absorbing, and with contrast typical for metal, and large (hundreds of microns), apparently representing hydroxyapatite, with a lower absorption coefficient. It should be noted that the microinclusions found in the control sample (Figure 1i,j) belong to the second type with a low absorption coefficient and likely have an organic origin (hard dental plaque). The average concentration of microparticles in samples of soft tissue biopsy specimens of granulation tissue obtained from bone bed areas in the region of four spontaneously disintegrated dental implants was ~2.4 × 10^−9^ µm^−3^. The concentration of microparticles in the control sample of granulation tissue was significantly lower than in the examined samples and equaled ~10^−10^ µm^−3^.

The results of the study of the elemental composition of the above bone bed biopsy samples by XRF are presented in Figure 2 (spectra were separated by the intensity axis for better perception). The interpretation of the obtained spectra revealed the presence of the following elements in the studied bone bed samples in the areas of self-disintegrated dental implants: Al, P, S, Ca, Ti, Cr, Fe, Ni, Cu, Zn, and Br (Figure 2, spectra 1–4). The control sample spectrum revealed the presence of the following elements in it: Ca, Fe, Zn, and Mn (Figure 2, spectrum 5). The presence of the Ar peak is explained by the fact that the measurements were performed in the air. It should be noted that, despite the absence of marked metallic inclusions, the control sample contains inclusions with low absorption coefficients, mostly of organic origin.

From the obtained X-ray studies, it can be assumed that the presence of metallic microparticles in the samples of soft tissue biopsy specimens of granulation tissue is the result of their emission from functioning dental implants, which are used to support orthopedic structures and/or direct from the structures themselves. This assumption became the basis for electron microscopy studies of the same samples to identify the emission objects and their elemental composition in comparison with the results of the X-ray studies.

### 2.2. Results of Electron Microscopic Studies

Figure 3 shows SEM images and corresponding energy-dispersive (ED) spectra obtained from the lower part of the Alpha Gate dental implant. It can be seen that the coating on the implant surface is not continuous, and there are areas with dark contrast up to several mm in size. ED analysis showed that the elemental composition in the areas with dark and light contrast is different. In the area with dark contrast, Ti prevails and there are also insignificant amounts of Fe, Ni, V, Ca, and Si (Table 1), while in the areas with light contrast, Ca prevails (Table 2).

The abutment study showed a predominance of Co and Cr elements, as well as the presence of small amounts of W, Fe, and S (Figure 4, Table 3 and Table 4). The dark areas where impairment of the coating is observed are characterized by increased S content.

The composition of the Alpha Gate system implants declared by the manufacturer is presented in [21]. According to this catalog, they consist of titanium alloy Ti_6_Al_4_VELI + bioactive CaP coating.

Thus, the methods of XMCT, XRF, and SEM identified impregnated metallic nano- and microscale particles in the structure of pathological tissue areas with the presence of a chronic inflammatory process (Figure 1, Figure 2, Figure 3 and Figure 4, Table 1, Table 2, Table 3 and Table 4). Their identity with the composition of the previously installed implants of the Alpha Gate system was revealed [21]. Therefore, it can be concluded that the detected metallic nano- and microscale particles have an emission property, and their presence in the pathological tissue areas adjacent to the implants is a consequence of the mechanical load during chewing.

### 2.3. Results of the Histological Study

Histological study of the tissue samples revealed that they are primarily represented by granulation tissue at different stages of maturation (Figure 5, position 2). On the periphery of the granulation tissue, there is dense, fibrous, often immature connective tissue (Figure 5, position 3). In some cases, we saw the inclusion of bone structures (Figure 5, position 1) represented by reticulofibrous and lamellar bone tissue. Thus, the histological results reliably agree with the X-ray tomography findings. Lympho- and plasmacytic infiltration, rare neutrophils, and histiocytic cells were found between connective tissue fibers and in granulation tissue. The majority of preparations also showed hemorrhages. In a number of cases on the inner surface of granulation tissue, there were foreign inclusions, presumably NSMP, representing single amorphous dark structures of non-histological nature (Figure 5, position 4).

### 2.4. Results of Dynamic Light Scattering (DLS) Studies

DLS was used to determine the parameters of nanoparticles in the supernatants in PBS for three different dilution ratios: The frequency of occurrence (kcps), particle size (nm), and polydispersity (%) (Table 5). As a result of the measurements of the three dilution ratios of the supernatants by DLS, samples were prepared to perform test experiments by visualizing cytometry. Based on the identity of the test experiment results, it was concluded that the maximum dilution of the initial supernatants could be used, i.e., 0.1 of the initial dilution ratio of 1.0. In earlier studies [22,23,24], as well as in this experiment, the particle size parameters did not exceed 220 nm, which was ensured by nanofiltration. On the other hand, the parameter of the frequency of occurrence of NSMP in this study increased ~5-fold, which allowed us to study its influence on the development of pathological reactions. The parameters for the Nobel Replace system (Danaher Corporation, Washington, DC, USA) were as follows: Particle size—88.16 nm; frequency of occurrence—36.6 kcps,; polydispersity—0.324%; for the Alpha Bio system (Alpha-Bio Tec Ltd., Petach Tikva, Israel), the particle size was 47.57 nm, frequency of occurrence was 34.4 kcps, and polydispersity was 0.434%.

### 2.5. Results of Visualizing Cytometry Studies

#### 2.5.1. Group 1

The results of the co-culture of Nobel Replace NSMP and THP-1 with induction by the donor serum with peri-implantitis (DSP) and lipopolysaccharide (LPS) are shown in Figure 6, Figure 7, Figure 8 and Figure 9.

#### 2.5.2. Group 2

The results of the co-culture of Alpha Bio NSMP and THP-1 with induction via the donor serum with peri-implantitis (DSP) and lipopolysaccharide (LPS) are shown in Figure 10, Figure 11, Figure 12 and Figure 13.

#### 2.5.3. Group 3 (Control)

The results in the control group when the THP-1 cell line was co-cultured with healthy donor serum (HDS, no previous dental implantation) with Nobel Replace NSMP and lipopolysaccharide (LPS) are shown in Figure 14, Figure 15 and Figure 16.

It should be noted that early-cell apoptosis could not be detected in this control subgroup. Late-cell apoptosis was observed immediately after their activation.

## 3. Discussion

Previously published articles [22,25,26] on in vitro experiments demonstrated the ability to increase the oxide layer of dental implants due to nanoscale particles under load, taking part in both physiological and pathological processes in bone tissue. The formation of a chronic immunopathological inflammatory process with an autoimmune component was shown on the basis of the critical supernatant dilution ratio. We assume that the mechanical overload of dental implants contributing to the formation of a critical dose of nanosized particles in tissues and their ability to coagulate is a paramount pathogenetic link in the development of immunopathological inflammation in the bone bed. The inability of immunocompetent cells to eliminate nanoparticles in a timely manner and their early death enables endogenous microflora to penetrate tissues and realize their pathogenicity, including through biofilm formation. Thus, clinically, in the form of mucositis, we see a manifestation of chronic immunopathological inflammation with an autoimmune component arising at the molecular level in the bone bed. The development of peri-implantitis, in this case, is the result of both bacterial and immunopathological inflammation due to the coagulation of nanoparticles to microparticles with the formation of granulation tissue as demonstrated in this article.

This article considers a clinical case of dental implantation in which the physical, histological, and immunological methods of research identified micro- and nanoscale particles as etiopathogenetic participants in the chronic inflammatory process in tissues in the area of previously installed dental implants with signs of pronounced peri-implantitis. The presence of nickel was detected in the studied tissues, which may indicate the possibility of a personalized delayed-type hypersensitivity reaction (DTH) in response to the microelement Ni2+ when it accumulates locally. Previously, the presence of inorganic inclusions was perceived as an artifact when performing histological studies of soft tissue biopsy specimens from granulation tissue of the bone bed. The present results clearly show that these inclusions correspond to the material of dental implants and/or orthopedic constructions in terms of structure and elemental composition.

On the basis of the results of X-ray and histological studies, it was concluded that there was an accumulative effect of NSMP and microparticles in the soft tissue components of the bone bed. These facts indicate the increased likelihood of the development of personalized inflammatory complications associated with individual sensitivity to metal alloy components.

It was suggested that nano- and microparticles may participate as an etiopathogenetic link in the occurrence of immunopathological inflammatory reactions, particularly locally in tissues. This served as the initiating aspect for a series of experimental studies on the role of metallic nano- and microparticles in the development of immunopathological inflammation in tissues.

It was found that when co-culturing NSMPs obtained from the surface of dental implants of two manufacturers’ systems, a decrease in early and late apoptosis was observed when DSP was added, in contrast to the subgroups with double signaling added at the expense of LPS. It should be noted that in the control group, with the addition of HDS to THP -1, in contrast to the subgroup with the presence of LPS inducing dual signaling on THP -1 from NSMP obtained with Nobel Replace, the absence of early apoptosis and enhancement of late apoptosis followed by cell necrosis were noted. The results obtained in the third subgroup of studies, in which the culture of Nobel Replace NSMP induced by DSP together with LPS was carried out, are worth special attention. Despite the presence of signaling from LPS to THP-1 cells, a decrease in late apoptosis was observed in both the group with the addition of Nobel Replace NSMP and the group with the addition of Alpha Bio NSMP. The resulting effect was noted in the properties of the donor serum with peri-implantitis to reduce the probability of late apoptosis and cell necrosis by co-culturing NSMP with THP-1.

In this case, imaging cytometry confirms the possibility of activation and death of immunocompetent cells during the joint cultivation of nanoscale metallic particles in the supernatants obtained from the surface of dental implants of two systems of manufacturers. An important fact is the identification of the influence of peri-implantitis patient serum on the reduction of early and late apoptosis of immunocompetent cells (THP-1—monocyte cell line), which was revealed here for the first time. This aspect was the basis for further genetic studies on the effect of nanoscale particles on the inhibition of early and late apoptosis to the necrosis of immunocompetent cells.

## 4. Materials and Methods

### 4.1. Clinical Case

Patient R. (male), 58 years old. The soft tissue biopsy specimens of the bone bed obtained from the oral cavity of a patient with clinical signs of chronic inflammation in the area of previously installed dental implants of the Alpha Gate system during surgical revisions, aimed at the treatment of peri-implantitis, were studied. During revisions in the area of self-disintegrated dental implants, surgical sanitation was performed with the removal of pathological tissues in four areas.

As a control, a granulation tissue sample obtained in the area of tooth 2.3 with a diagnosis of chronic periodontitis (ICD 10-K05.3), during flap surgery with guided bone regeneration on the upper jaw, was used.

### 4.2. Surgical Revision

The soft tissue biopsy specimens were immersed in 10% formalin and we subsequently studied both the X-ray tomography images and the elemental composition of the inclusions in all four specimens and the control. To independently confirm the presence of foreign particles impregnated into the structure of soft tissue biopsy specimens, an appropriate histological study was performed.

### 4.3. X-Ray Microtomography (XMCT)

The X-ray absorption microtomography method was used to detect foreign inclusions in the tissues of bone bed biopsy specimens impregnated from the surface of the oxide layer of medical devices and to determine their location, size, and concentration.

The measurements were performed on a “TOMAS” X-ray microtomograph [27]. A pyrolytic graphite monochromator crystal was used in these measurements (the beam size on the object is approximately 2 cm). The use of monochromatic radiation makes it possible to measure the real value of the linear attenuation coefficient of X-rays, which is important when investigating the elemental composition of samples. Measurements were performed using an X-ray tube with a molybdenum anode (energy 17.5 keV). Using this energy, the biological tissue, on the one hand, is transparent enough to maintain a high signal-to-noise ratio in the images, and, on the other hand, the boundaries of highly absorbing inclusions within this tissue are clearly visible.

The geometry of the experiments was as follows: Source–sample distance—1.2 m; sample–detector distance—0.02 m. With this experimental geometry, the radiation divergence is negligibly small, which is important for image analysis. Probing conditions: Accelerating voltage—40 kV; current—40 mA. To carry out tomographic studies, samples were rotated relative to a fixed vertical axis, and 400 projections were measured with a step of 0.5 degrees and an exposure of 4 s per frame. The XIMEA-xiRay11 high-resolution X-ray detector (XIMEA, Marianka, Slovakia) was used for the measurements. This detector allows for obtaining images with a resolution of 9 μm and a field of view of 36 by 24 mm.

The reconstruction for the case with strongly absorbing inclusions was performed using the algebraic method, which allowed us to reduce both the expression of “ray” artifacts in the reconstructed images and the variance of the linear absorption coefficient [28].

To evaluate the microinclusions contained in the samples, the following procedure for processing the obtained 3D images was applied. In the first stage, threshold filtering was applied in order to distinguish the boundaries of microinclusions. The threshold value was selected manually on the basis of analysis of the soft tissue absorption coefficient throughout the study. Further, the obtained binarized data were used to perform the labeling procedure. The application of such a procedure (segmentation) for all tomographic sections made it possible to isolate and analyze individual absorbing microinclusions, as well as to determine their geometric dimensions (equivalent diameter).

### 4.4. X-Ray Fluorescence Analysis (XRF)

The element composition of the soft tissue samples adjacent to the dental implant surfaces was determined by X-ray fluorescence analysis on an X-ray microtomograph using an X-123SDD detector-spectrometer (Amptek, Bedford, MA, USA). The radiation source was an X-ray tube with a molybdenum anode whose radiation was monochromatized by a single reflection from a highly perfect symmetric silicon crystal, with an orientation of (111). The radiation energy corresponded to the Kα1 molybdenum characteristic line and was 17.47 keV. The lower limit of measurements was limited by the parameters of the detector’s sensitive element and was ~1 keV. The energy resolution of the obtained fluorescence spectra was ~150 eV. The geometry of the experiments was as follows: A source–sample distance of approximately 1 m and a sample–detector distance of approximately 0.02 m. The size of the probing beam on the sample was ~1 mm horizontally and ~5 mm vertically. Probing conditions were as follows: Accelerating voltage—40 kV; current—40 mA. The exposure time was 600 s.

### 4.5. Scanning Electron Microscopy (SEM)

The surface structure and composition of the Alpha Gate system implants and orthopedic structure (abutment) were examined using a Scios Dual-Beam scanning microscope (ThermoFisher Scientific, Osiris, Waltham, MA, USA) in the secondary electron mode using energy-dispersive (ED) analysis.

### 4.6. Histological Studies

To increase the reliability of the results of physical study methods, duplicate histological studies of soft tissue biopsy specimens of the bone bed granulation tissue were performed.

A histological study of the tissue samples was performed immediately after the biopsy was obtained as a result of surgical treatment of peri-implantitis. The material was placed in a 10% solution of neutral formalin for 72 h, after which the tissue samples were washed in running water for 2 h. After standard histological preparation, the tissue samples were embedded in paraffin (Histomix, Biovitrum, St. Petersburg, Russia) using histological pouring rings (Biovitrum, St. Petersburg, Russia). Serial and semi-serial slices were made from the obtained blocks on a Microm microtome (from 3 to 7 μm). To reveal specific processes of connective tissue formation, the preparations were stained using the Mallory technique (Biovitrum, St. Petersburg, Russia).

### 4.7. Preparation of NSMP Supernatants from the Surfaces of Two Nobel Replace and Alpha Bio Dental Implant Systems

To obtain supernatants containing nanoscale metallic particles used in the experimental part of the study, 40 units of Nobel Replace and Alpha Bio dental implants were incubated in bidiistilled water for 5 days in a CO_2_ incubator. Nanoscale particles were obtained by the method described in the patent [29]. After incubation, tubes containing implants were treated with ultrasound at 35 kHz for 90 min to obtain highly concentrated supernatants with NSMP. Next, under laminar conditions, the implants were extracted from the tubes and the supernatants were combined in two tubes and diluted in phosphate-salt buffer solution (PBS), bringing the concentration to 1/20 of the original volume. A third tube of PBS was the control. The resulting supernatant samples were filtered through a Millipore syringe nanofilter (maximum pore diameter D = 0.22 μm). The purpose of increasing the ultrasound exposure time when obtaining the supernatants was to differentially diagnose the significance of the occurrence frequency and particle size parameters for cellular apoptosis.

### 4.8. Dynamic Light Scattering (DLS)

To identify nanoscale particles from the surfaces of two dental implant systems in the obtained supernatants and determine their size, a study was performed on a 90 Plus Partical Size Analyzer (Brookhaven Instruments Corporation, Holtsville, NY, USA) in multimodal mode using automatic 90Plus/BI-MAS and “dust cut-off” functions that allowed us to subtract very large objects, particularly dust. The filter value was 20. The measurements were recorded at a temperature of 25⁰C and a fixed light-scattering angle of 661 nm laser [22].

Using the DLS method, three parameters were measured: The NSMP diameter (D, nm), frequency of NSMP occurrence in the supernatants (ACR, kcps), and index of heterogeneity of NSMP distribution in supernatants—NSMP polydispersity (PD, %) [30]. Using the MS Excel software package ((Microsoft Office 2016), statistical data processing was performed for two systems, namely, Nobel Replace and Alpha-Bio.

### 4.9. Co-Cultivation of THP-1 and NSMP Obtained from the Surfaces of Two Nobel Replace and Alpha Bio Dental Implant Systems

THP-1 monocyte cells were cultured in RPMI 1640 medium (PanEco, Moscow, Russia) with the addition of 10% fetal bovine serum (FBS) (Biosera, Brittany, France), 2 mM L-Glutamine (PanEco, Moscow, Russia), NEAA (1x) (ThermoFisher Scientific, Gibco, Waltham, MA, USA), penicillin 100 units/mL (PanEco, Moscow, Russia), streptomycin 100 µg/mL (PanEco, Moscow, Russia), sodium pyruvate 1 mM (PanEco, Moscow, Russia), HEPES 10 mM (ThermoFisher Scientific, Gibco, Waltham, MA, USA), and 2 mM β-mercaptoethanol (Sigma-Aldrich, Burlington, MA, USA). Cells were poured into a 96-well plate, with 50,000 cells per well, in 100 μL of the medium. NSMP supernatants obtained from the surfaces of two dental implant systems with three different dilution ratios, namely, 0.1, 0.25, and 1.0 of the initial concentration, were added to the wells as stimulants. PBS was used as a negative control. These dilutions were used to estimate the threshold at which monocyte activation occurs. The nanoparticles diluted in 100 µL of medium were incubated for 5 min at room temperature with 50 µL of patient serum (without previously installed implants (HDS) or with peri-implantitis (DSP)) or with PBS and then added to the cells. In samples with lipopolysaccharide (LPS), 5 μL of LPS with a concentration of 10 μg/mL was added, and 5 μL of PBS was added to the remaining samples. The addition of lipopolysaccharide simulated the presence of Gram-negative bacteria as an additional signal from the antigen initiating bacterial presence under the conditions of the performed experiment. The final volume of the mixture was 255 μL. The cells were incubated for 24 h at 5% CO_2_ and 37⁰C before observation.

### 4.10. Methodology of Cell Preparation and Staining for the Analysis of Experimental Results by Visualizing Cytometry

Staining Procedure: Wash cells twice with cold Cell Staining Buffer and then resuspend cells in Annexin V Binding Buffer at a concentration of 0.25−1.0×10^7^ cells/mL; transfer 100 microl of the cell suspension to a 5 mL test tube; add 5 microl of Fluorescein isothiocyanate (FITS) Annexin V; add 10 microl of Propidium Iodid Solution; gently vortex the cells and incubate for 15 min at room temperature (25 °C) in the dark; add 400 microl of Annexin V Binding buffer to each tube; and analyze using flow cytometry with the appropriate machine settings.

### 4.11. Data Recording and Analysis on an ImageStream Mk II Flow Cytofluorimeter

Cell visualization and image recording were performed on an ImageStream Mk II imaging flow cytometer (IFC) using INSPIRE™ software (Luminex Corporation, Austin, Texas, USA). The IFC method is a flow cytometer with a set of 20 × −40 × −60 × lenses, 12 detection channels, and a Hamamatsu camera with a CCD matrix as a detector. The temporal resolution of the imaging flow cytometer camera allows up to 5000 images of cells per second, as well as the estimation of their fluorescence. A light-field image was used for direct light scattering, and a dark-field image was used as lateral light scattering. The fluorescence intensity was composed of the fluorescence intensity of each image pixel. All images were then characterized according to 85 parameters that included, in addition to traditional cytometry parameters, dozens of morphological characteristics that can be handled as conventional cytometric data. Images were recorded using a 488 laser at 50 microvolts with 60× magnification (camera lens inside the flow cell) and a low flow rate. At least 5000 events were recorded for each sample of cells with nanoparticles.

The obtained images were analyzed using IDEAS^®^ ImageStreamX software (Luminex Corporation, Austin, TX, USA).

### 4.12. Assessment Method for Early and Late Cell Apoptosis Using FITS Annexin V Apoptosis Detection Kit with PI

The FITC Annexin V Apoptosis Detection Kit with Propidium Iodid (PI) has been specifically designed for the identification of apoptotic and necrotic cells.

Annexin V (or Annexin A5) is a member of the annexin family of intracellular proteins that binds to phosphatidylserine (PS) in a calcium-dependent manner. PS is normally only found on the intracellular leaflet of the plasma membrane in healthy cells, but during early apoptosis, membrane asymmetry is lost, and PS translocates to the external leaflet. Fluorochrome-labeled Annexin V can then be used to specifically target and identify apoptotic cells. Annexin V Binding Buffer is recommended for use with Annexin V staining. Annexin V binding alone cannot differentiate between apoptotic and necrotic cells. To help distinguish between the necrotic and apoptotic cells, we recommend the use of our Propidium Iodid Solution (PI). Early apoptotic cells will exclude PI, while late-stage apoptotic cells and necrotic cells will stain positively, due to the passage of these dyes into the”nucl’us where they bind to *Deoxyribonucleic acid* (*DNA*). Propidium iodide is a fluorescent dye that binds to *DNA*. When excited by a 488 nm laser light, it can be detected within the PE/Texas Red R channel with a bandpass filter 610/10. It is commonly used in the evaluation of cell viability or *DNA* content in cell cycle analysis by flow cytometry [31,32,33,34].

### 4.13. Study Groups in the Investigation of the Role of Metallic Nano- and Microparticles in the Development of Immunopathological Inflammation in Tissues

In setting up the study, three groups were formed to study the role of immunocompetent cells as participants in the elimination of antigenic determinants in the form of nanoscale particles, the THP-1 monocytic cell line induced with human serum from a healthy donor (HDS) and serum from patient R. with a diagnosis of peri-implantitis (DSP): Group 1: Co-culture of Nobel Replace HSMP with THP-1 with induction of donor serum with peri-implantitis (DSP) and lipopolysaccharide (LPS); Group 2: Co-culture of Alpha Bio HSMP with THP-1 with induction of donor serum with a diagnosis of peri-implantitis (DSP) and lipopolysaccharide (LPS); Group 3—control: Co-culture of THP-1 cell line with serum from a healthy donor without prior dental implantation (HDS), with Nobel Replace NSMP and lipopolysaccharide (LPS).

## 5. Conclusions

Based on the results of these studies, it can be stated that nanoscale metallic particles can participate in both physiological and pathophysiological reactions of the immune system, i.e., be participants in the mechanisms of osseointegration and disintegration of dental implants. Their accumulation in tissues to critical values, called the “Critical Dose of Nanoscale Metallic Particles” (CDNanoMP), can contribute to the intensification of the inflammatory process in tissues locally, leading to the early death of immune system cells, which, in turn, is a trigger for the accumulation of nanoparticles against the background of chronic inflammation [35].

The development of mucositis and peri-implantitis may be related to the accumulation of nanoparticles as an initial trigger for aseptic inflammation in the bone bed. Failure of timely removal of nanoparticles locally in the tissues, due to early death of immunocompetent cells, may be the initiating aspect of immune-mediated inflammation with an autoimmune component. This state of cell reactivity seems to be secondarily aggravated by microbial invasion from the oral side, with the formation of biofilm.

Presumably, the qualitative and quantitative composition of antigenic determinants, namely, one-step signaling from microbiota penetrating the bone bed, damaged its own bone cells, and nanoscale metallic particles migrating from the surface of the dental implant oxide layer, increase the risk of fibroosteointegration, mucositis, and peri-implantitis after surgical intervention [36]. Based on the above, it can be concluded that the pathogenesis of mucositis and peri-implantitis should be correlated with the manifestations of endogenous infection in the oral cavity.

Due to these aspects of cellular and molecular features of tissue repair, in our opinion, delayed dental implantation is a more predictable method of treatment in terms of successful, long-term osseointegration of the dental implant, as opposed to immediate dental implantation with a single-stage load. The clinical manifestations of mucositis, in the case of initially successful osseointegration of dental implants, may be related to the development of a pathological inflammatory process directly in the bone tissue, which is the result of microparticle formation with the accumulating effect of nanoparticles due to mechanical loading and failure of timely elimination of antigenic determinants complex by immunocompetent cells. The performance of personalized studies on the analysis of the causes of complications in individual clinical cases provides an opportunity to determine the pathogenesis of the inflammatory process and substantiate the clinical symptomatology of the patient.

New physical, nanotechnological, immunological, and histological results of the studies presented above allow us to study cellular and molecular aspects of inflammation. The use of methods of X-ray microtomography, X-ray fluorescence analysis, and histological study of bone bed biopsy specimens in the area of dental implants makes it possible to prove the importance of nano- and microparticles’ involvement in pathological inflammatory processes, thereby substantiating the pathogenesis of their origin. The data obtained as a result of this research are the basis for the development of a test system that allows the prediction of complications associated with personalized sensitivity to the components of medical alloys.

## Figures and Tables

**Figure 1 ijms-24-02267-f001:**
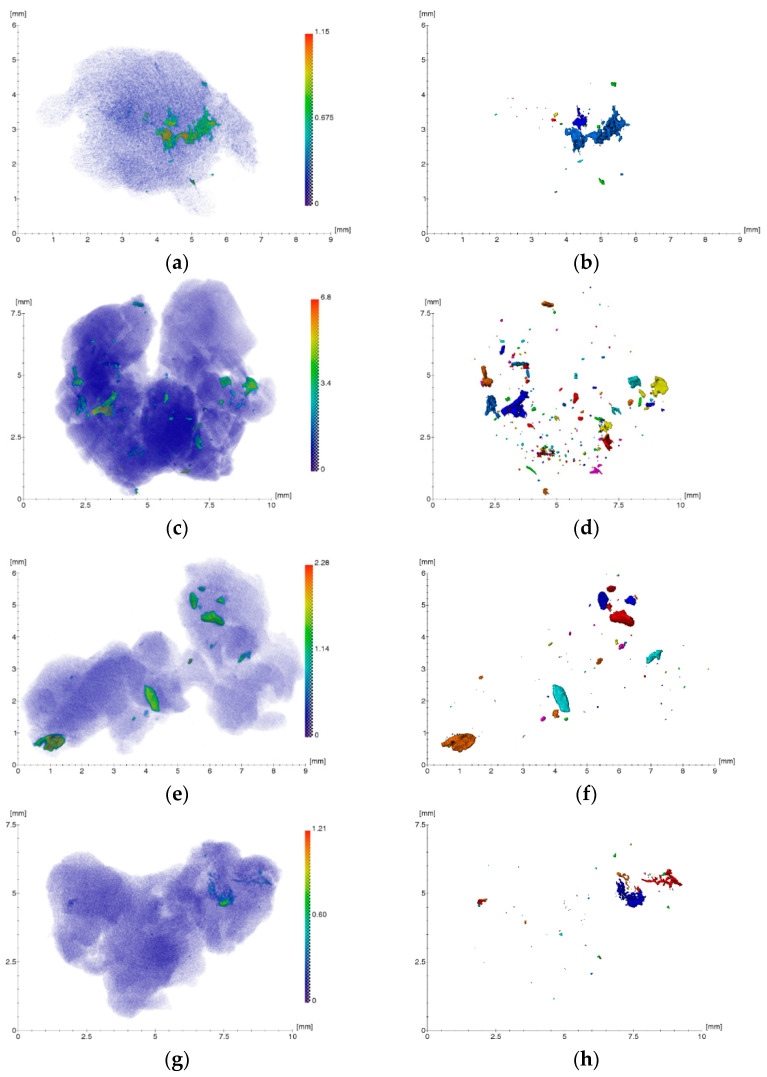
Results of three-dimensional reconstruction of biopsy specimens: (**a**,**c**,**e**,**g**) Original images of tissues with foreign inclusions; (**b**,**d**,**f**,**h**) resulting projections of foreign inclusions after segmentation procedure; (**i**,**j**) control specimen.

**Figure 2 ijms-24-02267-f002:**
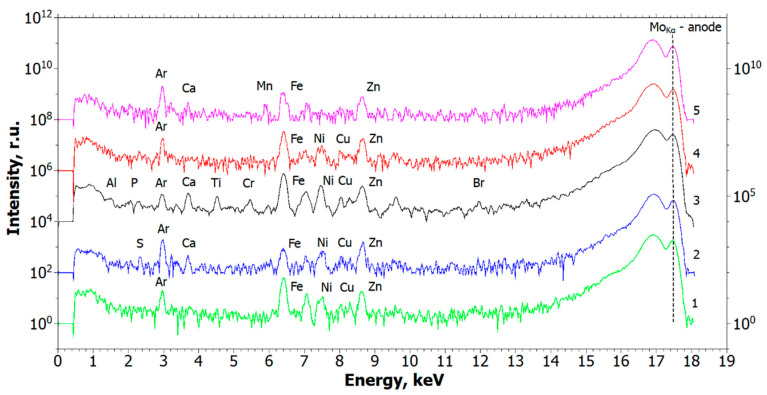
Spectra of samples elemental composition measurements by XRF: (1, 2, 3, 4) Bone bed zones in the area of four spontaneously disintegrated dental implants; (5) Control.

**Figure 3 ijms-24-02267-f003:**
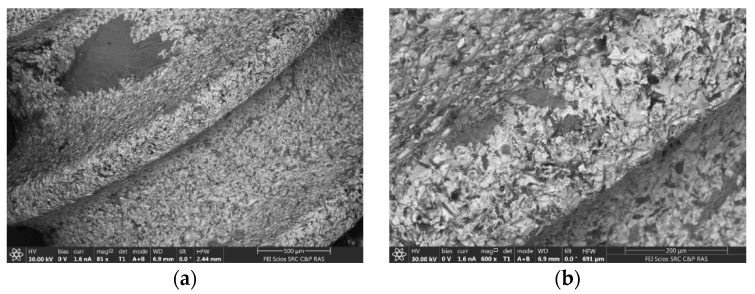
SEM images of the implant surface: (**a**) General view; (**b**) enlarged fragment; (**c**) general ED spectrum from the dark area; (**d**) enlarged section of ED spectrum from the dark area; (**e**) general ED spectrum from the light area.

**Figure 4 ijms-24-02267-f004:**
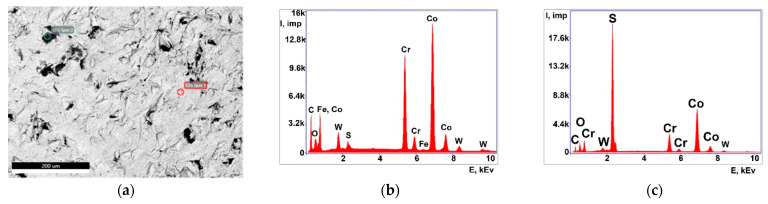
SEM images of the abutment surface: (**a**) General view; (**b**) ED spectrum from the light area; (**c**) ED spectrum from the dark area.

**Figure 5 ijms-24-02267-f005:**
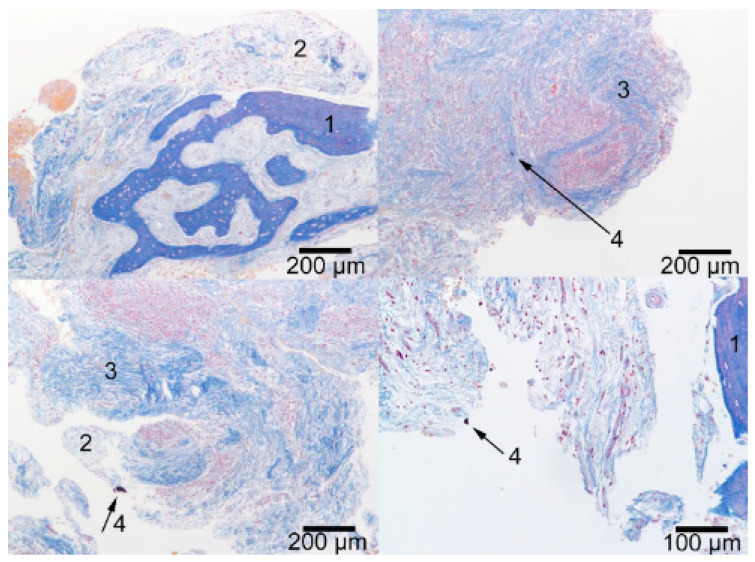
Results of histological study of biopsy specimens from the patient’s bone bed (Mallory staining): (1) Bone tissue; (2) Granulation tissue at different stages of maturation; (3) Maturing dense fibrous connective tissue; (4) Foreign particles.

**Figure 6 ijms-24-02267-f006:**
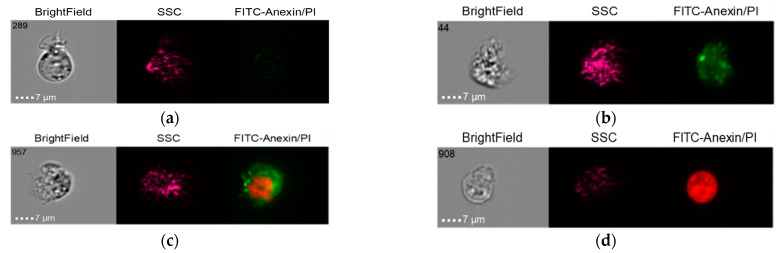
Live cells (**a**), early apoptosis (**b**), late apoptosis (**c**), necrosis (**d**) after co-culture of Nobel Replace NSMP with THP-1 cell line (Nobel Replace NSMP + DSP + THP-1).

**Figure 7 ijms-24-02267-f007:**
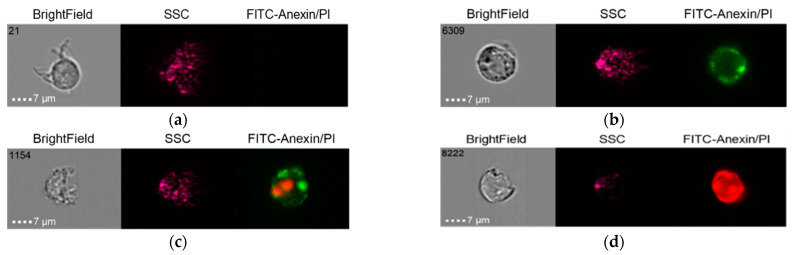
Live cells (**a**), early apoptosis (**b**), late apoptosis (**c**), necrosis (**d**) after co-culture of Nobel Replace NSMP with THP-1 cell line (Nobel Replace NSMP + LPS + THP-1).

**Figure 8 ijms-24-02267-f008:**
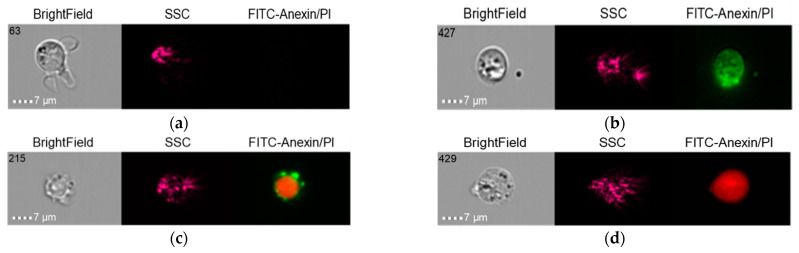
Live cells (**a**), early apoptosis (**b**), late apoptosis (**c**), necrosis (**d**) after co-culture of Nobel Replace NSMP with THP-1 cell line (Nobel Replace NSMP + DSP + LPS +THP-1).

**Figure 9 ijms-24-02267-f009:**
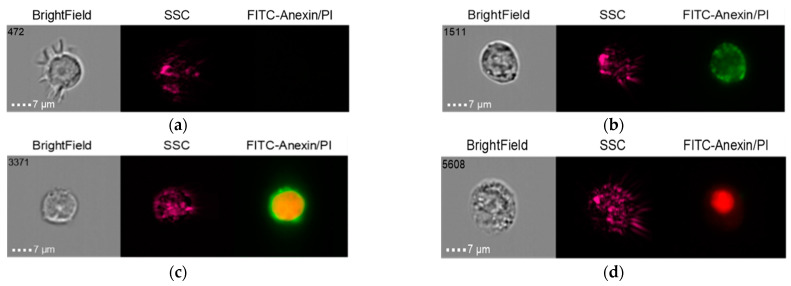
Live cells (**a**), early apoptosis (**b**), late apoptosis (**c**), necrosis (**d**) after co-culture of Nobel Replace NSMP with THP-1 cell line (Nobel Replace NSMP +THP-1).

**Figure 10 ijms-24-02267-f010:**
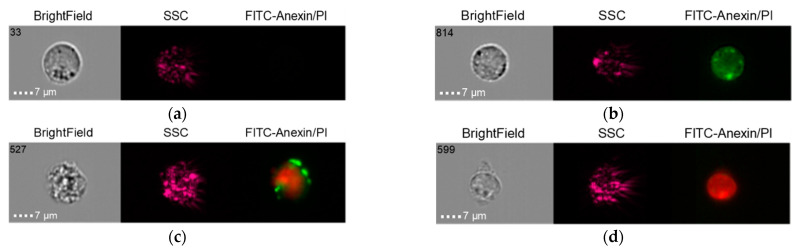
Live cells (**a**), early apoptosis (**b**), late apoptosis (**c**), necrosis (**d**) after co-culture of Alpha Bio NSMP with THP-1 cell line (Alpha Bio NSMP + DSP + THP-1).

**Figure 11 ijms-24-02267-f011:**
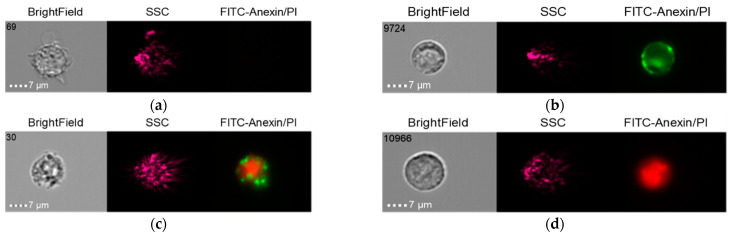
Live cells (**a**), early apoptosis (**b**), late apoptosis (**c**), necrosis (**d**) after co-culture of Alpha Bio NSMP with THP-1 cell line (Alpha Bio NSMP + LPS + THP-1).

**Figure 12 ijms-24-02267-f012:**
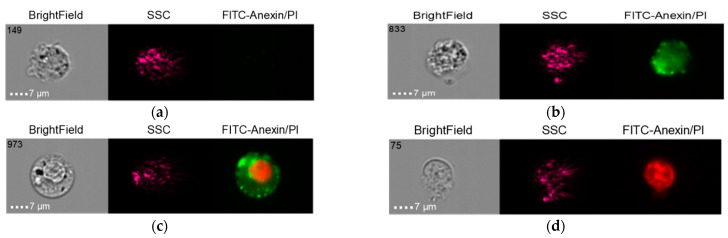
Live cells (**a**), early apoptosis (**b**), late apoptosis (**c**), necrosis (**d**) after co-culture of Alpha Bio NSMP with THP-1 cell line (Alpha Bio NSMP + DSP + LPS +THP-1).

**Figure 13 ijms-24-02267-f013:**
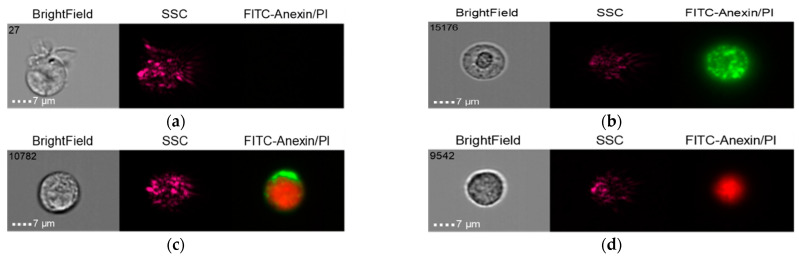
Live cells (**a**), early apoptosis (**b**), late apoptosis (**c**), necrosis (**d**) after co-culture of Alpha Bio NSMP with THP-1 cell line (Alpha Bio NSMP +THP-1).

**Figure 14 ijms-24-02267-f014:**
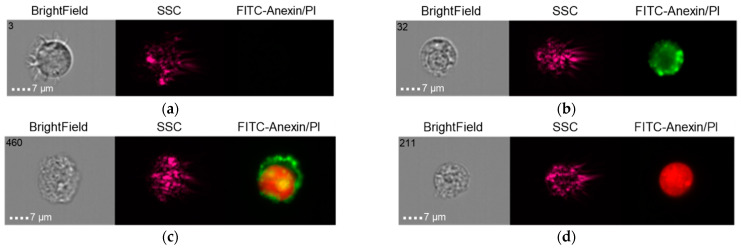
Live cells (**a**), early apoptosis (**b**), late apoptosis (**c**), necrosis (**d**) after co-culture of THP-1 cell line with healthy donor serum without previous dental implantation (HDS + THP-1).

**Figure 15 ijms-24-02267-f015:**
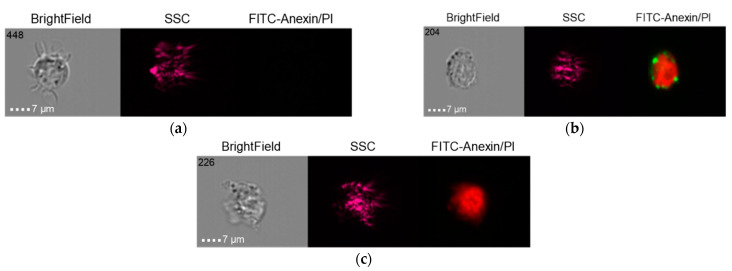
Live cells (**a**), late apoptosis (**b**), necrosis (**c**) after co-culture of THP-1 cell line with Nobel Replace NSMP induced by lipopolysaccharide (Nobel Replace NSMP + HDS + LPS + THP-1).

**Figure 16 ijms-24-02267-f016:**
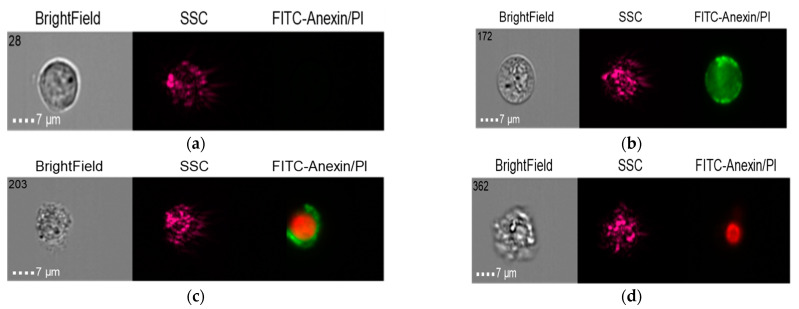
Live cells (**a**), early apoptosis (**b**), late apoptosis (**c**), necrosis (**d**) after co-culture of THP-1 cell line with Nobel Replace NSMP (Nobel Replace NSMP + THP-1+ HDS).

**Table 1 ijms-24-02267-t001:** Quantitative composition of one of the dark areas.

Element	Weight %	Atomic %
Na_K_	3.49	6.65
Al_K_	7.09	11.51
Si_K_	0.56	0.87
Ca_K_	0.04	0.05
Ti_K_	83.97	76.81
V_K_	4.26	3.66
Fe_K_	0.23	0.18
Ni_K_	0.36	0.27

**Table 2 ijms-24-02267-t002:** Quantitative composition of one of the light areas.

Element	Weight %	Atomic %
O_K_	13.38	26.41
As_L_	0.14	0.06
P_K_	23.83	24.30
Ca_K_	61.72	48.62
Ti_K_	0.93	0.61

**Table 3 ijms-24-02267-t003:** Quantitative composition of one of the light areas.

Element	Weight %	Atomic %
C_K_	27.96	54.24
N_K_	6.42	10.68
O_K_	8.28	12.05
Mg_K_	1.11	1.06
W_M_	4.65	0.59
S_K_	0.34	0.25
Cr_K_	16.55	7.41
Co_K_	34.70	13.72

**Table 4 ijms-24-02267-t004:** Quantitative composition of one of the dark areas.

Element	Weight %	Atomic %
S_K_	33.98	48.92
Ca_K_	0.17	0.20
Cr_K_	12.79	11.36
Co_K_	49.22	38.56
W_L_	3.84	0.97

**Table 5 ijms-24-02267-t005:** Parameters of NSMP contained in supernatants obtained from the surfaces of Nobel Replace and Alpha-Bio dental implants.

Sample	D, nm	ACR, kcps	PD, %
Nobel Replace 0.1_100	88.16	36.6	0.324
Nobel Replace 0.25_100	72.4	88.6	0.445
Nobel Replace 1_100	74.92	228.2	0.273
Alpha-Bio 0.1_100	47.57	34.4	0.434
Alpha-Bio 0.25_100	27.89	38.8	0.339
Alpha-Bio 1_100	13.74	148.3	0.328

## Data Availability

The data presented in this study are available on request from the corresponding author. The data are not publicly available because they are part of the patent application, they cannot be provided to a wide range of the professional community until it is received by the authors. In particular, the research results are part of the dissertation work that has not been defended. After the defense of the scientific qualification work, the data can be published in the public domain.

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
