# Peer review of "Cell-Molecular Interactions of Nano- and Microparticles in Dental Implantology"

_ijms, 2023, doi:10.3390/ijms24032267_

Round 1
Reviewer 1 Report
The author describes in the manuscript titled “Cell-molecular interactions of nano- and microparticles in dental implantology” a study on the role of metallic particles in the development of immunopathological inflammation in tissues. The article is well-written and demonstrates the possibility of involvement of nano and microparticles as an etiopathogenetic link in the onset of immunopathological inflammatory reactions. I recommend the publication of the manuscript after major revision:
1) SEM images demonstrate a high % of Ti, what could be the plausible effect of such a high percentage of Ti to the cell?
2) The author indicated that abutment study showed a predominance of Co and Cr elements. To what extent these percentage can affect the tissue area. All the nano- and micro particles would tend to diffuse further into tissue, what would be the degree of chronic inflammatory in such cases?
3) The author mentioned about inflammatory complications, what are the key complications or expected complications?
4) What should be the expected precautionary measure to be considered for dental implants?
Reviewer 2 Report
The topic is interesting and really up-to-date. However there the structure is not well-organized. The material and methods section should be included after introduction. I suggest also to present results and discussion in separate parts. Maybe the case report can be included as a part of discussion? The number of references is relevant to the subject of research.
Round 2
Reviewer 2 Report
The authors improved the manuscript according to the suggestions
